# The Case for Reading War Poetry as Ephemera

Julia Ribeiro S. C. Thomaz ⓘ

Centre des Sciencens des Littératures en Langue Française, Université Paris Nanterre, 92000 Nanterre, France; juliarscthomaz@gmail.com

**Abstract:** The First World War blurred the lines between "ordinary" and "literary" writing practices. Many sources corroborate this: necrologies written about poets who died in the act of writing not a poem but rather a letter, or introductions to poetry collections where bereaved families and friends admit they had no knowledge of their loved one's writing practices until they found a journal full of poems after the author's death, which they only published as a posthumous tribute. This article uses examples of French poetry of the Great War to explore this permeability between what is considered war poetry and what is considered war ephemera. The main question it addresses is what changes when we look at the war poems that were initially ephemera or ordinary writing. Whose stories get told when poetry is studied not as literature to be judged as accomplished or failed art but as a way of writing to make sense of the world? It argues that when we choose to read poems as ephemera and from the point of view of a larger anthropology of writing practices, diverse histories emerge and communities who write poetry not only as an artistic pursuit but also as a means of organizing experience and leaving traces behind reclaim ownership over their own narratives. This can challenge the false equivalence between the cultural history of warfare and an intellectual history of the elites at war and includes poetry within paradigmatic shifts that place objects at the centre of mediations of the experience of war.

**Keywords:** First World War; poetry; ephemera; France; *écriture ordinaire*; history; literature; anthropology

## 1. Introduction

*Recto*: a torn piece of paper, the letterhead indicating that it comes from Café Tortoni, a relatively unremarkable café in the French town of Nîmes (Figure 1). Verso: "*je soussigné déclare souscrire à Case d'Armons de Guillaume Apollinaire*" ["I declare that I would like to subscribe to Guillaume Apollinaire's *Case d'Armons*"] (Figure 2). One object, ordinary on one side but granting access to the war poems of one of the most famous French poets of the early 20th century on the other.

This piece of paper, sent by Guillaume Apollinaire from the frontline of the First World War to correspondents who could finance the publication of his war poetry collection (which was also written and published in the interval between battles and on whatever paper he could find at the front—Figure 3), embodies the present article's hypothesis: the Great War blurred the lines between "ordinary" and "literary" writing practices.

Indeed, the violent rupture with normality represented by armed conflict sheds light on a diffuse and non-literary relationship with poetry, which goes beyond the intellectual elites and the horizon of the literary field. In other words, it is possible that poetry has always *done* and will always *do* things that go beyond literature and literary value, but warfare blurs the lines between this practice and literary poetry, shedding light on poems that were written as simple traces of experience. There is, therefore, a lot to be gained from reading war poetry as ephemera, understood here in the terms used by the War Ephemera project on their website: "any small physical traces of everyday life other than printed books" (War Ephemera-Home n.d.). The final exclusion is interesting, especially given that, like Apollinaire's *Case d'Armons*, many war poems were later published in collections. This

study, therefore, presupposes a shift in perspective: it requires acknowledgement of the fact that, before they became books (which were often published posthumously by bereaved family and friends), some poems were simply traces of everyday life in the trenches and later became literary objects due to their ties with the conflict and the readers' avidness for war literature. When war poetry is stripped of its exclusively literary aura and is read from the point of view of a larger anthropology of wartime writing practices, diverse histories emerge, and communities (especially rural and labourer ones) who write poetry not only as an artistic pursuit but also as a means of organizing experience, leaving traces behind and bestowing sense upon the world, reclaim ownership over their war narratives. This article, therefore, explores the permeability between what is considered poetry (and literature more broadly) and war ephemera. The main questions it addresses are what changes when we look at the war poems through the prism of their initial stages as ephemera or ordinary writing? Whose stories get told when poetry is studied not as literature to be judged as accomplished or failed art but as a way of writing to make sense of the world and to create understanding by leaving traces?

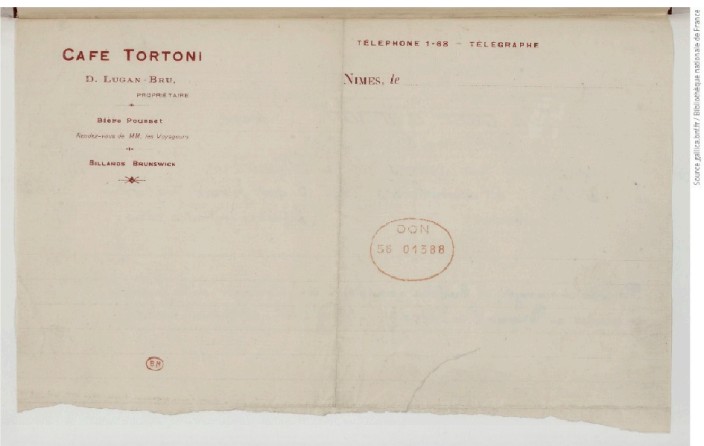

**Figure 1.** Recto of one of the *"Bulletin de souscription à la Case d'Armons"*, held in the *Bibliothéque national de France* and digitised via *Gallica*. It indicates that Apollinaire kept the headed paper of the café he used to go to during his time in the barracks at Nîmes and took it with him to the front, where it was then repurposed.

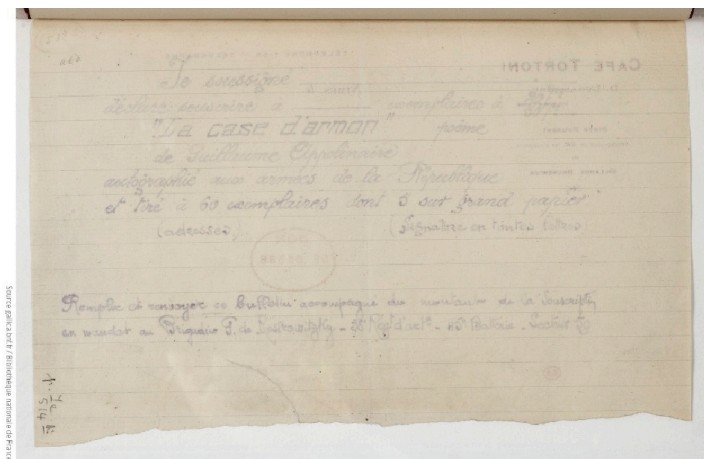

**Figure 2.** Verso of the same subscription bulletin. It reads "Je soussigné déclare souscrire à la Case d'Armons de Guillaume Apollinaire, autographié aux armées de la République et tiré à 60 exemplaires, dont 5 sur grand papier. (adresses) (signature en toute lettre). Remplir et renvoyer ce Bulletin accompagné du montant de la souscription en mandat au Brigadier G. de Kostrowitzky. 38e Regt. D'Artillerie, 45e Batterie- secteur 49".

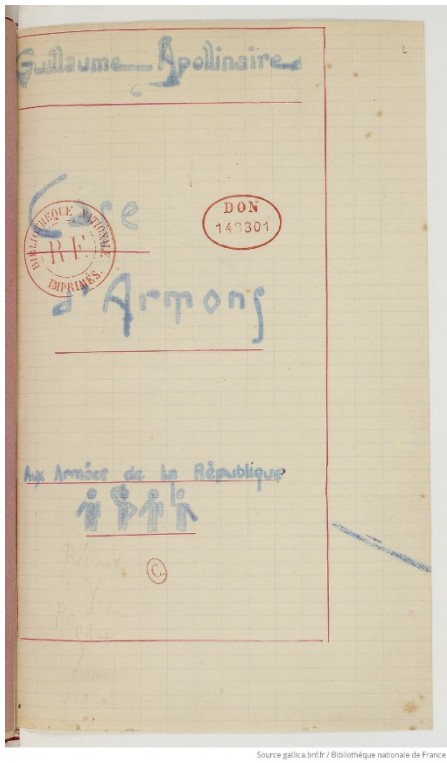

**Figure 3.** Front cover of one of the 60 copies of *Case d'Armons*, which was handwritten and then reproduced using the poet's regiment's roneo duplicator in 1915. This copy is kept at the *Bibliothèque nationale de France* and has been digitised via *Gallica*.

To answer these questions, the present article will use sources relating to French poetry of the First World War. Most of these sources were collated through the digital humanities project Poésie Grande Guerre, an online relational database that links individuals socially identified as poets, their poetic production, and their war experience. Indeed, the French case lends itself particularly well to this kind of study since French poetry of the First World War has been excluded from literary periodization, from historiography, and from collective memory (Ribeiro S. C. Thomaz 2019). Stripping war poetry back to its "trace" element would be much harder if the case study were the British corpus, where a canon (embodied mainly by Wilfred Owen and Siegfried Sassoon) that does not represent the full diversity of experiences of the Great War has come to dominate the collective imagination. France, on the other hand, offers a "blank canvas" (even though poetry was published, circulated and valued as a document and as literary production during the war) where, instead of a canon, poetry can be read as a cultural practice that is not limited to the intellectual elites that circulated in the literary field. The absence of an existing canon of poetic memory in France allows for a convergence between History and Literature mediated not via Sociology (as has previously been the case in France) but rather via Anthropology, interrogating poetic cultures and writing practices during the First World War. This study, therefore, uses poetic sources, especially those written at the frontline, such as Apollinaire's *Case d'armons*, but also other sources that gravitate around this poetic corpus, particularly the poets' necrologies published in the *Bulletin des écrivains combattants* (a monthly newspaper sent free of charge to writers at the frontline) and the anthologies commemorating poets who died in combat.

The study is not completely new. Laurence Campa, who is best known for her research about the rare French poets who, exceptionally, escaped the silence surrounding the poetry of the First World War, has already studied one of the privileged mediums for permeability between war poetry and war ephemera: the postcard. She demonstrates how, during the First World War, postcards became a poetic space where the circumstance of

the conflict was represented, absorbed, and reinvented (Campa 2022). By going beyond the traditional dichotomy between written and visual supports, Campa places the texts as artifacts, restituting poetry to its materiality: a poem–postcard demonstrates that poetry is a trace used to mediate and make sense of the experience of war not only to oneself but also to civilian contemporaries and future generations.

The phenomenon has also been studied in the context of the Second World War, for both writing in general and for poetry in Yiddish. In his examination of writing practices of those "condemned to death" (meaning living in ghettos or concentration camps) under the Nazi regime, Michel Borwicz highlights a general and spontaneous phenomenon where victims (regardless of their level of formal education) left messages for those who would survive the war—a true "graphomania" (Borwicz 1996, p. 38). Immediately after the German invasion of Poland, the force of the events turned thousands of people into "authors" (Borwicz 1996, p. 58), and as the war became a genocide, small poetic brochures were not written and edited for aesthetic pleasure but rather to act as a last will and testament (Borwicz 1996, p. 56–57), demonstrating that forms historians or literary historians would consider literary (a series of poems collected in brochures) were actually serving as traces—as ephemera. This relationship between poetry and ephemera during the Second World War is accentuated when this poetry is written in Yiddish, and the trace left behind is the language itself, which needs to be saved from extermination even if the people who speak it will not. Rachel Ertel claims that this Yiddish corpus prompts researchers to examine the links between poetry and history, poeticism and historicity (Ertel 1993, p. 10). The author comes to two interesting conclusions. On the one hand, Yiddish poetry of the Shoah embodies the paradoxical nature of war writing (whether "literary" or "ephemeral"): it is impossible to write, but it is also impossible not to write (due to social and circumstantial impositions which, as we will see below, characterize "ordinary writing"). Ertel's second conclusion is even more relevant for the purposes of the present article: the author claims that an inversion of analysis is needed to understand the phenomenon of war poetry. When speech (*parole*) is powerless to leave traces of a horrible reality behind, mediated (poetic) language is the only thing capable of signifying this reality. While more research needs to be conducted comparing the poetic corpora of both world wars, the hypothesis that extreme violence pushes all written traces towards poetic mediation makes a compelling case for the study of war poetry as war ephemera that may or may not be rendered perennial by publication but whose initial vocation is not to become a work of literature.

This precedent of approaching poems through the prism of ephemera and traces, and therefore, through their materiality, allows for a new relationship between History and poetry, triangulated via Anthropology, which can rely on studies about "ordinary" writing practices and of poetry as part of larger processes of "making" (Ingold 2013). The former is an expression popularized by French anthropologist Daniel Fabre (1993) to describe the writing gesture (and its products) that has no desire to "*faire oeuvre*" (become a work of art) or to be printed, published, and therefore, consecrated as literature. Indeed, Fabre claims that, whether associated with intense collective moments or with a simple routine, these "*écritures ordinaires*" (ordinary writings) have only one purpose: to leave a trace behind. While most literate people do leave traces in their everyday lives, many of them, according to Fabre, would have a negative answer to the questions "do you write?" or "are you a writer?"[1] This points towards a common-sense dichotomy between writing practices that require an engagement of the self (and which would warrant a positive answer to the question, turning individuals into "people who write") and a second type of writing practice, exterior to the self, socially or circumstantially imposed. But Fabre himself indicates that this gap should be breached, mentioning early 20th-century wartime "*cahiers de chansons*" (notebooks where soldiers would copy song lyrics to ensure memorization—a practice that echoes the ordinary and socially imposed practice of learning by copying in French schools) that were usually treated and kept as books, acquiring the "sacred" status as any other printed and literary object would.

This article will continue Fabre's movement towards breaching the gap between "ordinary" and "literary" writing, demonstrating how war poetry can be read as both. It will examine several case studies that show how French poetry of the First World War transitioned from ephemera to literature (especially for poets who died in combat) and then back to ephemera in a process of "demotion" that inscribes itself in the wider exclusion of this corpus described above. In its second section, the article will examine what is gained from reading war poetry from the point of view of this permeability: how it allows researchers to bypass matters of canonicity and, therefore, of opposition between History and poetry; how it challenges the idea that cultural history is circumscribed to intellectual elites; how it restitutes the materiality of texts which, along form and content, becomes an element of analysis; and, finally, how, by doing all of this, it allows diverse histories to emerge.

## 2. Poetry as Ephemera—How?

In France, the First World War began with a poem, at least for the officers who graduated from the École spéciale militaire de Saint-Cyr the day before the general mobilization was decreed. This poem was not supposed to leave traces. During his end-of-year speech, which marked the graduation of the Montmirail class of 1912–1914, Jean Allard-Méeus recited his own poem "*Demain*" ["Tomorrow"]. Referencing the 1870–71 Franco-Prussian war throughout the whole piece, Allard-Méeus promises that the soldiers from Alsace and Lorraine (who were then in the German army) would soon rejoin their brothers in arms under the French flag. To the Germans, Allard-Méeus issues a warning: be mindful of your country because the French army will soon take it. In a letter to his mother, where he recounts his graduation dinner, Allard-Méeus writes: "*[. . .] puis, au milieu de l'émotion grandissante, j'ai dit 'Demain'. [. . .] Jamais, ma petite maman, je ne dirai plus ces vers, car jamais plus je ne serai á la veille d'un jour de départ pour là-bas, au milieu de mille jeunes gens tremblant de fièvre, d'orgueil et de haine. J'ai sans doute trouvé dans mon émoi personnel l'accent qu'il fallait avoir, car j'ai fini mes vers au milieu d'un frisson général.*" ["then, amidst growing emotion, I said 'Demain'. [. . .] Never, my little mummy, will I say these verses again, because never again will I relive the eve of a departure to go over there, surrounded by a thousand young people trembling with fever, pride and hatred. I have surely found, in my personal emotion, the right tone, because I finished my verses to general chills"]. The choice of verbs is an interesting one: Allard-Méeus considers that his verses were meant to be "said", not written or read. These verses are the fruit of a powerful emotional response to a once-in-a-lifetime situation. Allard-Méeus takes the notion of ephemera beyond the one adopted by the War Ephemera project and in this article: this poem is so ephemeral it was never meant to constitute a trace—it was only supposed to be alluded to in letters and in the journals and memoirs of the other members of the Montmirail class, existing only as a memory. Yet someone, whether Allard-Méeus himself or one of his classmates, wrote the poem down, preserving it. Allard-Méeus was hit by two bullets (allegedly one in his head and one in his heart—symbolic wounds that kill not only the man but also the intellect and emotions that make him a poet) near Verdun, on the 22nd of August 1914, less that one month after delivering his famous speech. In 1920, "*Demain*" completed its transition from the most ephemeral of poems into literature, integrating the collection *Rêves d'amour! Rêves de Gloire*, published by a bereaved mother who mourned not only Allard-Méeus but his father, also killed in combat (Allard-Méeus 1920).

This example raises the hypothesis that the First World War (or perhaps any moment of violent rupture with ordinary life) not only creates new poets but also renders perennial and tangible, previously diffuse, non-literary ways of engaging with poetry that perhaps had always been present in the French society but which the war turned into literary objects. In other words, perhaps writing poems for speeches, sending verses back home in letters, and composing odes to the people of the past that were never meant to be read was a widespread cultural practice, but because industrial warfare killed so many of these "ordinary poets", it generated an influx of mausoleum-books filled with poems never intended for publication.

However, this transition from ephemera to literature triggers a backlash. In the interwar years, when the veteran movement was marked by pacifist rhetoric (Prost 2014), patriotic wartime poetry, such as Allard-Méeus's, became associated with civilian propaganda (with poets of the home front being accused of glorifying a war they knew nothing about and this criticism being extended to war poetry in general), and war poetry fell out of taste. In a way, it was, therefore, "demoted" to the status of ephemera once more, mere traces of a circumstance that were used in a wider propagandistic effort. In the words of Ian Higgins, who published an anthology of French poetry of the First World War, these poems were perceived as "purveyors of outdated jingoistic clichés written in doggerel" (Higgins 1996, p. vii). Because some of the poems published during or immediately after the Great War were initially ephemera, the public and literary critics denied the whole practice the status of "literature", failing to see that this corpus sits between "ordinary" and "literary" writing practices. Meanwhile, those interested in literature as documents deemed poetry too "literary" to be taken seriously. Jean Vic, for example, defends the exclusion from his manual *La littérature de guerre* (published between 1918 and 1923) of self-published brochures with little literary interest (Vic 1918, 1923). Meanwhile, Jean Norton Cru, who wanted to compile a list of the best testimonies of the First World War, claimed that poetry was so "literary" that its acceptance as a historical source posed more inconveniences than advantages (Norton Cru [1928] 1993, p. 11). While the process of ephemera becoming literature and then becoming ephemera again seems to imply a certain hierarchy between ephemeral and published poetry, Allard-Méeus's case should, this article argues, be read as doing the exact opposite: it demonstrates how war renders the frontier between literature and ephemera more fluid, that war poetry should be understood as a practice occupying this liminal space and, therefore, encourages the refusal of any hierarchy between both spheres.

While Allard-Méeus's poem, letter, and subsequent posthumous collection are, therefore, very interesting and important, they hardly help diverse histories emerge. Jean Allard-Méeus, his father, was a cavalry lieutenant whose son seemed destined to receive officer training at Saint Cyr, an education often reserved for the elites. The poet also received the French Legion of Honour after his death. Having been one of the instigators of the early-war phenomenon of Saint Cyr students going into combat wearing their military school uniforms, including the class markers of the white gloves and sabres, Allard-Méeus was, in many ways, a typical member of the French upper and middle classes, for whom officer training was often an inherited right. The transition between war ephemera and war poetry was not, however, restricted to the elite, and this is illustrated by the case of the *Chanson de Craonne*, the most famous French protest war song of the Great War, which entered the quasi-canon of the period (since it is impossible to speak of a canon of First World War poetry in France) after it had a "definitive version" fixated by the communist poet Paul Vaillant Couturier, in 1934. During the war, it was circulated at the front both orally (in the form of a popular *chanson*) and possibly as a written note passed from soldier to soldier or sent home in letters (which reminds us of Daniel Fabre's argument about the *cahiers de chanson* also occupying a liminal space between ephemera and literature). The first trace of the words registered by historians comes from a letter by the soldier Jean Duchesne, who had fought at the Somme and the Chemin des Dames, sent to his wife on the 17th of February 1917 (proving, therefore, that the Chanson de Craonne was already popular before the 1917 mutinies during which it was believed to have been composed):

> *Je te dirai que je tenvois la chanson des embu[s]qués et tout se que je te prise sait de la conservait car sait la seule chaonson qui me pai et elle est raielle du reste tu pourra la profondire de toi-même tu vaira que sai raielle et aussi tot reçu raiecri moi pour que je sui sur que tu lait car sa mennuirai quel soit perdu et dit moi si ele te plai* (Marival 2014, p. 32). [I will tell you that I am sending you the song of the embusqués (draft evaders) and all I ask is that you keep it because it is the only song I like and it is real furthermore you will be able to deepen yourself you'll see that it is real and

as soon as you receive it write back to me because it would annoy me if it got lost and tell me if you like it].

The English translation can hardly express how the letter's orthography deviates from standard written French (lack of punctuation, spelling and conjugation errors, and so forth), demonstrating that the French Third Republic's mandatory education instituted in 1882 contributed to increased literacy levels and to grant formal French precedence over regional dialects; in fact, the ways in which people engaged with the French language were diverse. It is also an indication that Jean Duchesne belonged to a non-dominant class, possibly coming from a rural background. The *Chanson de Craonne* itself, perhaps due to the fact that it was published by Vaillant Couturier in 1934 in *Commune* (the official journal of the Association of Revolutionary Writers and Artists), went down in history as an artistic expression of the class struggle that animated the 1917 mutinies. Regardless of its social undertones, however, one thing draws our attention in this letter, which is particularly interesting in the context of the permeability between war ephemera and war poetry: Duchesne's concern about the preservation of the song. While his orthography demonstrates that he belongs to what has been named the *scripteurs peu letters* (semi-literate writers—Dal Bo 2021), he still saw writing as a means to ensure his favourite song would leave a trace. In 1934, the poet Paul Vaillant Couturier, whose relationship to the French language and to the act of writing was in many ways opposite to Duchesne's, did the same and published the song's lyrics merely as a text (without the accompanying melody), effectively dissociating it from the *chanson* genre and turning it into the most famous communist poem of the First World War but also ensuring it would survive. Like Jean Allard-Méeus's "*Demain*", the history of the *Chanson de Craonne* demonstrates the fluidity between war poetry and war ephemera and shows that the war crystallised in writing diffuse poetic practices that were largely based on orality. Jean Duchesne's letter and the two parallel roads the *Chanson de Craonne* took towards its preservation in writing show, however, that exploring this poetry as ephemera can help diverse histories (of semi-literate and rural populations, for example) come to light, demonstrating that the choice of writing poetry in order to leave traces (poetry as ephemera) cut across class distinctions and different levels of formal education.

One of the best sources to identify this permeability between poetic and ephemera writing during the First World War is probably the necrologies of poets who died in combat. A close analysis of these texts shows that many of those socially designated as poets and as war poets, moreover, never had the chance to write about their war experience or even to publish poems before the conflict broke out, either because of their age or because of their social origins. Obituaries that pay homage to the poet instead of to the poems, to the man instead of the work, allow survivors to also grieve for all the poems that could have been written, and a quantitative analysis of the list of combatant poets included in the *Poésie Grande Guerre* database (https://pgg.parisnanterre.fr, accessed on 29 April 2024 (more details about the database's construction can be found in Ribeiro S. C. Thomaz 2019)) indicates that around 40% of them were "war poets without war poems". This finding will be discussed in the section below as one of the positive results of reading war poetry and war ephemera as integrating the same spectrum of wartime writing practices, but for now, it is interesting to examine how these obituaries promoted the transition between war poetry and war ephemera as part of their construction of the myth of the war poet. This is part of a larger movement that presents the First World War combatants as a lost generation whose greatest accomplishments will never come to be, and it requires the attribution of poetic characteristics to ephemera written by combatants, a poeticism which hints at the poems that died with the should-have-been poet. Therefore, the *Anthologie des Écrivains Morts à la Guerre* tells how Louis Dulholm-Noguès died "as a poet" because he died "as a writer", which means writing—even if what he was writing was not a poem. The evidence of this poetic death is a letter to his mother, unfinished because the poet was hit by a shell during combat in Marcilly (Battle of the Marne):

*Ma chère Maman, Après bien de voyages, bien de péripéties que je vous raconterai tout au long si Dieu me permet de revenir, je vous écris sous une pluie d'obus et de mitraille. Déjà dans ma section des braves et des bons amis sont tombés. Mais il paraît que les Prussiens se font terriblement ramasser. Et cela console des pertes douloureuses que nos éprouvons. Êtes-vous toujours à Paris ? Je crois que les boches ne peuvent. . .* [My dearest Mummy, After many travels, many adventures which I will tell you about in due course if God allows me to come back, as I'm writing under a storm of shells and machine gun fire. In my section, brave and good friends have already fallen. But it seems that the Prussians are being terribly beaten. And that comforts us in the painful losses we are having. Are you still in Paris? I think the Boches cannot. . .]

The ellipsis at the end of the letter is reinterpreted by Dulholm-Noguès's friend, who found it on his body, and later by the obituary's author as a representation of the poet's unfinished work. Jules Dupin's journal is also cited in his obituary in the *Anthologie*. A daily practice initiated when the poet was only a teenager, the journal opens with the line (later interpreted as prophetic): "*Le devoir avant tout, le devoir c'est tout ce qu'on doit faire*" ["Duty above all, duty is all that which must be done"]. The journal's final line, which the obituary's author describes as weakly scribbled in pencil (a sign of the poet's fleeting life), reads: "*J'ai l'angoisse de mourir. Des obus tombent sur nous. Mon Dieu, pitié, mon Dieu!*" ["I'm afraid to die. Shells are falling around us. My God, have pity, my God!"]. According to Georges Dessoudeix, the author of his brother in arms's obituary, all of Dupin's spirit is contained in these pages, starting with duty and ending with God. The necrology also indicates that some poets' efforts to maintain a daily writing habit, even if "just" of letters and journals (ephemera), and points towards a tenacity of writing practices that became synonymous with the myth of the war poet. The mourning of the unfinished or even unwritten work and the attribution of poeticism to letters and journals, which transforms every poet at war into a war poet, also completes a transmutation of war ephemera into war poetry: everything written by a poet is assumed to be poetic.

Of course this transmutation is simply a metaphorical one, and it was questioned after the war (as illustrated by Jean Vic's contempt towards self-published books cited above) (Vic 1923), but also during the conflict itself. Just because the heroic circumstances of a poet's death were enough to grant verses (which were often not yet ready for publication) or even ephemera the status of a work of art composed by someone socially accepted as a war poet, that does not mean that all or even most readers were willing to accept what was later considered "bad" poetry simply because the author had died in combat. In the eleventh number of the *Bulletin des écrivains de 14–15* (later the *Bulletin de Écrivains Combattants*), dated September 1915, Henry Guilbeaux, a pacifist and anarchist poet who had been exempted from all military obligations, wrote ironically about war literature:

*Il est constant que lorsqu'un écrivain meurt, les lauriers et les louanges lui soient généreusement offerts. On le dote de talents et de vertus. On lui accorde avec une rare magnificence tout ce qu'avec obstination on lui avait refusé de son vivant. La guerre, qui renverse les valeurs, bouscule les conventions, crée une nouvelle morale et de neuves coutumes, et transforme toute l'atmosphère, a fait surgir des écrivains et des héros. Quiconque avait écrit naguère quelque sonnet et mourut sur le champ de bataille est célébré, fêté, et le Bulletin des Écrivains publié à Paris et hors commerce, nous révèle quantité de littérateurs dont jusqu'à ce jour personne n'avait entendu prononcer le nom.* [It is a constant that when a writer dies, he receives laurels and praises generously offered. We bestow upon him talents and virtues. We grant him, with a rare magnificence, everything we had obstinately denied him when he was alive. War, which inverts values, pushes conventions, creates a new moral and new customs, and transforms the whole atmosphere, brings forth both writers and heroes. Whoever previously wrote a sonnet and died on the battlefield is celebrated, and the Bulletin des Écrivains, published in Paris and free of charge, reveals a great number of littérateurs[2] whose names nobody had heard before].

The debate about how war shed light on literature whose poor quality would not have been accepted in ordinary times begins, therefore, relatively early. In other words, the war granted certain works a literary status they were perceived as not deserving, earned due to their authors' combat exploits. This was one of the many reasons for the exclusion of French war poetry from literary periodization and collective memory during the interwar years. Indeed, because war ephemera and previously critically rejected poetry were so easily transformed into war poetry, with the legitimacy this title grants, during the war, especially for poets who died in combat, when the circumstances changed, the public assumed that all war poetry was bad and had simply been published because the poets had been combatants. Therefore, war poetry, as a category, became once again associated with war ephemera: a mere trace of the past, which can be finally seen for what it is (allegedly "bad" poetry) once the weight of the event is lifted and the main preoccupations can be with "good" literature once more.

The examples provided above are far from exhaustive and offer only a glimpse at the diverse writing practices which compose French poetry of the First World War. Guillaume Apollinaire, for example, who was considered a poet before the war and became a war poet, shows that the war constrained even the most famous of poets to material difficulties, weaving war poetry and war ephemera into the same fabric. The examples are, however, indicative of the trend this article has sought to explore: a permeability between war poetry and war ephemera, with authors and texts transitioning between these two categories at different points in time. This indicates that poetry, during the First World War but maybe even before, accomplished things that went beyond literature and were closer to ephemera and ordinary writing and that "composing a poem" and "leaving a trace behind" are not opposite acts, as the distinction between "literature" and "ephemera" would have us perceive them but rather a spectrum on which there are many grey zones and points of transition. The next section will examine why seeing war poetry and war ephemera as permeable can shed light on aspects of the First World War thus far ignored and change the way war writing is perceived.

## 3. Poetry as Ephemera—Why?

More important, however, than these examples that demonstrate how war poetry and war ephemera were fluid categories is the heuristic value of this fluidity and what it teaches us not only about the poetry of the First World War but also about the value of ephemera. There is much to gain from reading war poetry and war ephemera as belonging to the same spectrum of wartime writing practices, primarily for a reconciliation between History and Literary Studies. These disciplines have been at odds for both French and British literature of 1914–1918. Indeed, Ann-Marie Einhaus (2011) identifies not one but rather two canons of the Great War, attributed to the different expectations of historians and literary critics. While, according to Einhaus, the First World War is a perfect example of an event whose commemorations are founded on a convergence between History and Literature (and this is true even for France, where prose seems to dominate), canons stills pose problems for a dialogue between the disciplines, since each wants to establish competing hierarchies for the texts:

> When we talk about the literature of the First World War, it is important to be aware of two distinct bodies of texts. If we talk of a canon of First World War literature, we generally tend to refer to a canon that is not 'literary' in a strictly formal and aesthetic sense, but informed by socio-cultural interests. While a literary canon of early twentieth-century writing that incorporates the literature of the war is interested primarily in how a given text expresses thoughts, concepts or ideologies, the canon of First World War literature is interested primarily in what is expressed. (Einhaus 2011, s.p.)

Reading war poetry as war ephemera and vice versa, paying particular attention to the moments in which a text transitions from one to the other and to the shared cultural practices that unite rather than divide different kinds of authors and writings can help bypass the (often stalling) questions of literary quality or historical representativeness,

finally offering a common ground where Literature and History can meet. In other words, a text's "patrimonial" and canonical status becomes part of the phenomena historical and literary analyses try to explain instead of a determining factor for which texts are studied or not. If we recognize that "great" literature and "representative" ephemera are not set categories and often share the initial stages of mark-making and trace-leaving, we can grasp the broader dimensions of writing practices that neither the historical canon nor the literary one can view independently.

While this offers an advantage from the point of view of First World War studies and for scholars who seek truly interdisciplinary research, the implications go beyond that. Reading war poetry as war ephemera can help challenge the idea that cultural history is one that should be reserved for the intellectual elites, as some of the French war poets have been studied ([Mariot 2013](#)). If war poetry and war ephemera can seamlessly transition between one another, then the traces left behind by people who were not professional poets before the war or who did not become professional poets after it are just as legitimate expressions of individuals confronted with mass violence. The transition between war poetry and war ephemera turns every contemporary of the war into a lyrical subject and every poet into a witness. Moreover, it is representative of how the contemporaries themselves saw wartime writing, as indicated by the obituaries mourning poetry that could have been or that is hidden in "ordinary" pieces. The ease of the transition between war poetry and war ephemera, therefore, indicates that the war either pushed "ordinary" writers towards poetry because the genre was potentially perceived as the best one to sublimate the circumstances or, an equally valid hypothesis, because the war softened the borders (and, therefore, the autonomy) of the literary field, bringing to the surface diffuse poetic practices that are sometimes closer to ephemera than to literature. In both cases, reading war poetry as war ephemera points towards the need for a cultural history and a poetic history, more specifically, that is not restricted to the urban intellectual elites and to the traditional literary and editorial or book history since this approach sheds light on diverse ways of composing and sharing poems.

Reading war poetry and war ephemera as belonging to the same spectrum of practices and as constantly transforming into each other also raises the question of materiality. On the one hand, as we have seen with the case of Guillaume Apollinaire, the proximity between war ephemera and war poetry sheds light on the material constraints informing the latter. It can also help explain war poetry's very existence as a widespread wartime practice: it is easier to write a poem than a novel in the context of rationed paper since, as we saw with *Case d'armons*, a poem can be scribbled on any spare sheet. Moreover, it is also easier to compose a poem in your head and memorize it even when you are in a situation where you cannot write it down, just as it is easier to share a poem (orally or through ephemera, as was the case of the *Chanson de Craonne*) to ensure that the trace survives even if the poet dies. Reading war poetry as ephemera, therefore, goes beyond the traditional questions of "what" a poem says about the war or "how" it says it: the "why" and the "on what"/"under which conditions" become just as important in this analysis that goes beyond the text itself and focuses on poetry as a cultural practice. On the other hand, the relationship with war ephemera also has the power to free poetry from materiality altogether and to shed light on the actors' views of "war poetry" as a social category that does not depend on the texts themselves. As we have seen from the examples of a "poetic essence" being bestowed upon letters and journals, many of the people recognized as war poets had never written about the war or had never even written poems. If historians and literary historians insisted on seeing war poetry and war ephemera as separate, they would potentially ignore the phenomenon of the creation of war poets without war poems. The transmutation between war poetry and war ephemera is, therefore, a central element in establishing a relationship between the literary and material cultures of the war, whether to unite them or to dissociate them altogether.

The most important change that happens when war poetry and war ephemera are observed together is, however, the diverse histories that can emerge from this shift in per-

spective. Roland Barthes (1966) argued that narrative was transhistorical and transcultural and had an ability to unite people of different or even opposing cultures, whereas poetry is dependent on the readers' cultural level. This led to an association of any historiography that takes into account poetic sources as a historiography of writings whose authors shared a culture, an education, and a certain sense of subjectivity, to use Spivak's (1988) term with the historians themselves. Thus, even in cases when the history being written had no bearing on colonialism (indeed, the poetry of the French colonial troops remains to this day largely unexplored), poetic histories that fail to see what war poetry and war ephemera have in common still constitute a sort of epistemic violence (Spivak 1988). In other words, they reduce to silence and to the role of the subaltern (or at least to the role of "bad poets") any actors who wrote poetry for the sake of leaving a trace behind simply to conform to rural and working-class poetic traditions. These histories have, for example, dissociated First World War poetic practices from peasant and labourer poems, which had been a part of the proletarian experience for centuries before the war and which were brought to the trenches by rural and working-class soldiers (Ribard 2018). Seeing war poetry as ephemera restores to these communities the survival through mastery (Smith 2007, p. 60) they so actively sought, recognizing the traces they left behind and, thus, acknowledging their stable identities. In her study of the poetry written and shared by workers from as early as the 17th century, Dinah Ribard (2018) claims that, in addition to its traditional literary functions, poetry is a social practice that also fulfils non-literary roles for those writing and reading it, especially with regards to the transmission of labour-related knowledge. The influence of this industry-based poetic tradition is seen in Theodore Botrel's poem "*Le Soldat chante*" ("The Soldier Sings") (Botrel 1914), which amalgamates poetry and popular songs and claims that soldiers sing both to learn their new wartime occupation and to distract themselves from their labours. This poetic connection between the trenches and the factories and fields reintroduces the question of class into the cultural history of the First World War, and it can only emerge if poetry is seen as having non-literary, ephemeral properties.

If reading war poetry as ephemera can help historians see the non-literary functions of poetic practices and the complex relationship between poetry and materiality, then another diverse history emerges in addition to that of working-class soldiers: the poetic history of colonial troops. Mostly oral, this corpus has not been the object of a systematic study precisely because it is dissociated from traditional literary modes of expressing the war and even from written ephemera. However, if, like any war poetry, this corpus is approached through the bias of ephemera and trace-leaving, new sources emerge. Laurent Jolly (2018, p. 79) transcribed a poem inspired by the Somali tradition of oral pastoral and brought back from the trenches by camel herder-turned-mercenary Djama Haïd, who transmitted it as an oral poem to his sons, interviewed by Jolly in 2012. Another poem, by a Muslim French colonial trooper who wished for a victory for the Sultan (and, therefore, a victory for the Ottoman Empire and the Allied Powers), was intercepted by the postal control in Marseille and later discussed by Jean Yves Le Naour (2018, p. 308). These two examples show how the poetry produced by colonial soldiers was as intimately linked with ephemera as that produced by soldiers of the metropole, if not more. With regards to the poetry of both British and French colonial troops, Toby Garfitt (2017, p. 48) concludes that "even those associated with the British and French empires did not primarily express themselves in English or French, and the majority of them not on paper. [...] We are reminded of how painfully narrow our habitual perspective of war poetry is". This perspective can be broadened if war poetry is considered in terms of what unites it with war ephemera instead of what distances them. If poetry is no longer read as a work of art but as a trace left behind, the North African French oral poems and chants recorded by German linguists in prisoner camps, for example, can be seen as "war poetry" and integrate the poetic history of the war, as can the examples selected by Jolly and Le Naour. This, in turn, can broaden our view of what a poem actually is, making diverse histories emerge.

## 4. Conclusions

In conclusion, the First World War, in general, and the French case, in particular, due to its lack of a pre-established canon, are of particular interest for interrogating the fluid boundaries between war poetry and war ephemera. The Jules Ferry laws of 1881 and 1882, which made education free and universal in France, finally allowed oral poetic practices to leave written traces. Furthermore, the war, during which writing was often a soldier's only link with home and, metonymically, with every aspect of his life that was not related to the conflict, amplified the writing. The legitimacy granted to those who fought and died in the war then introduced this "poetic ephemera" (both poems unintended for publication and ephemera to which a certain poetic character was attributed) into the literary field. In other words, a close examination of French poetry of the First World War demonstrates how permeable the categories of "war poetry" and "war ephemera" actually are. This, in turn, brings about epistemological changes that are fundamental for History, Literary Studies, and for the relationship between these two disciplines. Reading war poetry as another trace left behind, as another piece of war ephemera, does risk oversight of the relationship between a poet, whether amateur or professional, and the expectations of the literary field. It also risks ignoring the importance of poetic models, especially the national canon taught in schools, in shaping poetic responses to the war. On the other hand, reading war poetry as ephemera can broaden our very idea of what is war poetry and, more importantly, of who gets to be considered a war poet and, therefore, to master their own story by telling it. Given the diverse histories that emerge if we allow ourselves to read war poetry as yet another trace left behind, a temporary blindness to the literary function of this corpus (which has already been studied to a greater or lesser extent) is a price we should be willing to pay.

**Funding:** This research received funding from the Prix d'Études des Mondes Contemporains (2020) and the Bourse Gerda Henkel–Historial de la Grande Guerre (2021).

**Institutional Review Board Statement:** Not applicable.

**Informed Consent Statement:** Not applicable.

**Data Availability Statement:** Data supporting these results can be found in the *Poésie Grande Guerre* relational database, available on https://pgg.parisnanterre.fr, accessed on 29 April 2024.

**Conflicts of Interest:** The author declares no conflict of interest.

## Notes

[1] It is interesting to note that this is the very definition of poetry given by First World War poet Jean Cocteau in his 1949 film *Orphée*. When the protagonist, a poet, is questioned by judges in a courtroom about his occupation: "*Qu'appelez-vous poète?*" ("What do you mean by poet?"), he replies "*Écrire, sans être écrivain*" ("To write, without being a writer").

[2] A pejorative term used to designate people who write what they think is literature but who do not have enough quality to warrant the designation of "writer".

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
