# Peer review of "The Case for Reading War Poetry as Ephemera"

_genealogy, doi:10.3390/genealogy8020055_

Round 1

Reviewer 1 Report

Comments and Suggestions for Authors

This well-written and cogently argued article offers a fascinating reflection on the processes by which poems begin as ephemera and become literature (and sometimes vice versa), and the role that the heightened social and cultural pressures of war can play in these processes. The author offers a clearly structured argument and uses well-chosen examples, arguing convincingly that the French context is particularly well suited to the discussion given the absence of an established literary canon of war poetry (though many of the observations made are just as valid in the Anglophone context, and undoubtedly others). The article argues convincingly that the reading of war poetry as originating in ephemeral forms enables us to view handwritten ephemera such as letters or diaries and either initially or permanently unpublished war poetry or song lyrics as part of a broader cultural practice in wartime that carries over from peacetime, but is polarised by the combatant status of the writers. The author also makes a strong case for the potential of understanding ephemera and war poetry as part of the same continuum to achieve a greater inclusivity of the category of war poetry, by making space for rural and industrial working-class writers as well as writing by colonial soldiers. The argument draws helpfully on theoretical approaches to writing practice from social anthropology, and ultimately seeks to bridge a conceptual gap between historians’ and literary critics’ engagement with war poetry. The interdisciplinary approach adopted here is exemplary and very helpful for thinking about the wider significance of non-canonical war writing in particular. If anything, I felt an even more emphatic case could be made about short literary and ephemeral forms as outlets for the overlooked and disenfranchised – a link that is articulated most strongly in comparison with the ephemera and writings produced by Holocaust victims and survivors. My only two suggestions for minor revisions are, first, a thorough copyedit that the author is no doubt planning to undertake anyway, and second, a quick consideration of Tim Kendall’s decision to include a section on soldiers’ songs in his OUP anthology Poetry of the First World War (2013), which could add a useful comparative element to the discussion of the movement of the ‘Chanson de Craonne’ from song to poetry.

Author Response

Dear reviewer,

Thank you for your valuable feedback. I have conducted a thorough proof-read of the paper as requested. Thank you for drawing my attention to Tim Kendall's inclusion of songs in his anthology, which I now address in my paragraph about the Chanson de Craonne. I have also written another sentence about poetry's power as an outlet for the disenfranchised, and I agree with your suggestion about the importance of emphasising this point once more.

Reviewer 2 Report

Comments and Suggestions for Authors

The case for reading war poetry as ephemera

This is an interesting study that seeks to establish categories of verse-writing issuing from the experience of the First World War as either ‘poetry’ or ‘ephemera’, based largely on the way it is later processed by those who inherit it in one way or another. The author goes on to argue that this distinction is not only an unreal one, but also actually harmful, and politically undesirable, in its devaluing of ‘ephemeral’ verse as merely a crude record of experience, and presumably (though this is not quite clear) its enshrining ‘poetry’ as an aesthetic object with little value as historical witness.

The work of already established writers—Apollinaire is cited—could immediately be considered, and consumed, as poetry. But verse written, often in the form of private journals and letters, by thousands of ordinary combatants must have been lost or destroyed, unless the friends or family of the writer put it into print. In such cases, usually patriotic verses after a few years fell out of critical favour, and so were often reassigned to the category of ephemera. The case is made that there remains a value in such writing, and that the distinction between ‘war poetry’ and ‘war ephemera’ is a pernicious one.

I found the use of the word(s) ‘ephemera(l) confusing.  Some poems are ‘simply traces of everyday life in the trenches’, without literary ambition, and some people ‘write poetry not only as an artistic pursuit but also as a means of organizing experience’ (3).  At many points (the essay is somewhat repetitious) the author seems to regard ‘ephemera’ and ‘traces’ as synonymous. We are told such writings ‘were actually serving as traces – as ephemera’ (4), and that it will be helpful to approach them ‘through the prism of ephemera and traces’ (5). But in ordinary language these words are almost opposite in meaning: ‘ephemera’ normally connotes transitoriness, while traces are what survives.

What changes when the poetry is read as communicative and existential ‘ephemera’, rather than as Art—or vice versa? Actually, there is very little in this essay about the actual practice of reading. The author’s main point is to show that there is a permeability between poetic and ephemeral writing. ‘If we recognise that “great” literature and “representative” ephemera are not set categories and often share the initial stages of mark-making and trace-leaving, we can grasp the broader dimensions of writing practices that neither the historical canon nor the literary one can view independently.’ (10) We need not confine one set of writing to the Faculty of Letters and the other to the Faculty of History.

I think the case is made persuasively. I am a bit less convinced that, these days, it is a very controversial one. Perhaps a further agenda is revealed in the claim that ‘Reading war poetry as war ephemera can help challenge the idea that cultural history is one that should be reserved to the intellectual elites’ (10). Who actually subscribes to this idea?

If ‘ephemeral’ means tied to a particular circumstance or occasion, addressed to a particular person, raising issues arising from everyday experience, then surely there can be no doubt that ephemeral writing can be continuous with poetry. One of the best witnesses of the permeability of the ‘poetic’ and ‘ephemeral’ is none other than Apollinaire himself, the Apollinaire of Alcools as much as of Case d’Armons.

Author Response

Dear reviewer,

Thank you for your valuable feedback. My article states that I use the War Ephemera project's definition of ephemera: "traces of everyday life other than printed books", but I agree with you that, due to the terms being seemingly antonymous in ordinary language, I needed to clarify that I would be using them interchangeably because of that definition. I have now changed the paragraph to read: ". There is therefore a lot to be gained from reading war poetry as ephemera, understood here in the terms used by the War Ephemera project on their website: “any small physical traces of everyday life other than printed books”. Despite being seemingly contradictory in ordinary language (since “ephemera” connotes something transitory while traces are what survives), ephemera will thus be used in this article as traces of the war, and the two terms will be used interchangeably. Poetry is therefore close to ephemera in the sense that they are both traces of everyday life at war which have survived, and whenever poetry is not published in a printed book it becomes ephemera. The final exclusion in the War Ephemera definition (printed books) is interesting, especially given that, like Apollinaire’s Case d’Armons, many war poems were later published in collections." 
Thank you for drawing my attention to the importance of a work on the reception and reading of this poetry, I look forward to working on that in a future piece.